# Psychological and pharmacological interventions for posttraumatic stress disorder and comorbid mental health problems following complex traumatic events: Systematic review and component network meta-analysis

Peter A. Coventry[1,2]*, Nick Meader[1], Hollie Melton[1], Melanie Temple[3], Holly Dale[4], Kath Wright[1], Marylène Cloitre[5,6], Thanos Karatzias[7], Jonathan Bisson[8], Neil P. Roberts[8,9], Jennifer V. E. Brown[1,2], Corrado Barbui[10], Rachel Churchill[1], Karina Lovell[11], Dean McMillan[2,12], Simon Gilbody[2,12]

1 Centre for Reviews and Dissemination, University of York, York, United Kingdom, 2 Department of Health Sciences, University of York, York, United Kingdom, 3 Schoen Clinic, York, United Kingdom, 4 School of Health Sciences, University of Manchester, Manchester, United Kingdom, 5 National Center for PTSD Dissemination and Training Division, VA Palo Alto Health Care, Menlo Park, California, United States of America, 6 Department of Psychiatry and Behavioral Sciences, Stanford University, Stanford, California, United States of America, 7 Edinburgh Napier University, School of Health & Social Care, Edinburgh, United Kingdom, 8 Cardiff University, School of Medicine, Cardiff, United Kingdom, 9 Cardiff and Vale University Health Board, Cardiff, United Kingdom, 10 Department of Neurosciences, Biomedicine and Movement Sciences, University of Verona, Verona, Italy, 11 Division of Nursing, Midwifery and Social Work, University of Manchester, Manchester, United Kingdom, 12 Hull York Medical School, University of York, York, United Kingdom

* peter.coventry@york.ac.uk

## Abstract

### Background

Complex traumatic events associated with armed conflict, forcible displacement, childhood sexual abuse, and domestic violence are increasingly prevalent. People exposed to complex traumatic events are at risk of not only posttraumatic stress disorder (PTSD) but also other mental health comorbidities. Whereas evidence-based psychological and pharmacological treatments are effective for single-event PTSD, it is not known if people who have experienced complex traumatic events can benefit and tolerate these commonly available treatments. Furthermore, it is not known which components of psychological interventions are most effective for managing PTSD in this population. We performed a systematic review and component network meta-analysis to assess the effectiveness of psychological and pharmacological interventions for managing mental health problems in people exposed to complex traumatic events.

### Methods and findings

We searched CINAHL, Cochrane Central Register of Controlled Trials, EMBASE, International Pharmaceutical Abstracts, MEDLINE, Published International Literature on Traumatic

**Data Availability Statement:** Data are available from the primary research papers, which are listed in the references.

**Funding:** Funding was received from the UK National Institute for Health Research (NIHR) Health Technology Assessment programme (ref: 16/11/03) (to PAC) (https://fundingawards.nihr.ac.uk/award/16/11/03). The funders had no role in study design, data collection and analysis, decision to publish, or preparation of the manuscript.

**Competing interests:** The authors have declared that no competing interests exist.

Stress, PsycINFO, and Science Citation Index for randomised controlled trials (RCTs) and non-RCTs of psychological and pharmacological treatments for PTSD symptoms in people exposed to complex traumatic events, published up to 25 October 2019. We adopted a non-diagnostic approach and included studies of adults who have experienced complex trauma. Complex-trauma subgroups included veterans; childhood sexual abuse; war-affected; refugees; and domestic violence. The primary outcome was reduction in PTSD symptoms. Secondary outcomes were depressive and anxiety symptoms, quality of life, sleep quality, and positive and negative affect. We included 116 studies, of which 50 were conducted in hospital settings, 24 were delivered in community settings, seven were delivered in military clinics for veterans or active military personnel, five were conducted in refugee camps, four used remote delivery via web-based or telephone platforms, four were conducted in specialist trauma clinics, two were delivered in home settings, and two were delivered in primary care clinics; clinical setting was not reported in 17 studies. Ninety-four RCTs, for a total of 6,158 participants, were included in meta-analyses across the primary and secondary outcomes; 18 RCTs for a total of 933 participants were included in the component network meta-analysis. The mean age of participants in the included RCTs was $42.6 \pm 9.3$ years, and 42% were male. Nine non-RCTs were included. The mean age of participants in the non-RCTs was $40.6 \pm 9.4$ years, and 47% were male. The average length of follow-up across all included studies at posttreatment for the primary outcome was 11.5 weeks. The pairwise meta-analysis showed that psychological interventions reduce PTSD symptoms more than inactive control (k = 46; $n = 3,389$; standardised mean difference [SMD] = −0.82, 95% confidence interval [CI] −1.02 to −0.63) and active control (k-9; $n = 662$; SMD = −0.35, 95% CI −0.56 to −0.14) at posttreatment and also compared with inactive control at 6-month follow-up (k = 10; $n = 738$; SMD = −0.45, 95% CI −0.82 to −0.08). Psychological interventions reduced depressive symptoms (k = 31; $n = 2,075$; SMD = −0.87, 95% CI −1.11 to −0.63; $I^2 = 82.7\%$, $p = 0.000$) and anxiety (k = 15; $n = 1,395$; SMD = −1.03, 95% CI −1.44 to −0.61; $p = 0.000$) at posttreatment compared with inactive control. Sleep quality was significantly improved at posttreatment by psychological interventions compared with inactive control (k = 3; $n = 111$; SMD = −1.00, 95% CI −1.49 to −0.51; $p = 0.245$). There were no significant differences between psychological interventions and inactive control group at posttreatment for quality of life (k = 6; $n = 401$; SMD = 0.33, 95% CI −0.01 to 0.66; $p = 0.021$). Antipsychotic medicine (k = 5; $n = 364$; SMD = –0.45; –0.85 to –0.05; $p = 0.085$) and prazosin (k = 3; $n = 110$; SMD = −0.52; –1.03 to −0.02; $p = 0.182$) were effective in reducing PTSD symptoms. Phase-based psychological interventions that included skills-based strategies along with trauma-focused strategies were the most promising interventions for emotional dysregulation and interpersonal problems. Compared with pharmacological interventions, we observed that psychological interventions were associated with greater reductions in PTSD and depression symptoms and improved sleep quality. Sensitivity analysis showed that psychological interventions were acceptable with lower dropout, even in studies rated at low risk of attrition bias. Trauma-focused psychological interventions were superior to non-trauma-focused interventions across trauma subgroups for PTSD symptoms, but effects among veterans and war-affected populations were significantly reduced. The network meta-analysis showed that multicomponent interventions that included cognitive restructuring and imaginal exposure were the most effective for reducing PTSD symptoms (k = 17; $n = 1,077$; mean difference = −37.95, 95% CI −60.84 to −15.16). Our use of a non-diagnostic inclusion strategy

may have overlooked certain complex-trauma populations with severe and enduring mental health comorbidities. Additionally, the relative contribution of skills-based intervention components was not feasibly evaluated in the network meta-analysis.

## Conclusions

In this systematic review and meta-analysis, we observed that trauma-focused psychological interventions are effective for managing mental health problems and comorbidities in people exposed to complex trauma. Multicomponent interventions, which can include phase-based approaches, were the most effective treatment package for managing PTSD in complex trauma. Establishing optimal ways to deliver multicomponent psychological interventions for people exposed to complex traumatic events is a research and clinical priority.

## Author summary

### Why was the study done?

- Complex traumatic events are of a multiple or prolonged nature and are increasingly prevalent owing to unprecedented levels of population displacement, armed conflict, and increased recognition of childhood sexual abuse and domestic violence.

- People exposed to complex traumatic events are at risk of not only posttraumatic stress disorder (PTSD) but also other mental health problems.

- There are evidence-based psychological and pharmacological treatments for single-event PTSD, but it is not known if people who have experienced complex traumatic events can benefit and tolerate commonly available treatments.

- To inform treatment guidelines and future research, a broad evidence synthesis is needed that goes beyond existing knowledge to identify candidate interventions for mental health problems associated with complex trauma.

### What did the researchers do and find?

- We undertook a systematic review and meta-analysis of the effectiveness and acceptability of psychological and pharmacological treatments for mental health problems in veterans, refugees, victims of childhood sexual abuse and domestic violence, and war-affected populations.

- We used network meta-analysis to disentangle the relative contribution of different components of psychological treatments.

- The meta-analysis showed that psychological treatments are effective for treating PTSD, anxiety, and depression and improving sleep in people with a history of complex traumatic events.

- Pharmacological interventions were less effective than psychological interventions for treating PTSD symptoms and improving sleep.

- Trauma-focused treatments were the most effective approaches, but these treatments tended to be less effective in veterans and war-affected populations.

- Multicomponent interventions that included two or more components were the most effective for treating PTSD symptoms, and these approaches were promising for the management of disturbances of self-organisation.

### What do these findings mean?

- Existing evidence-based trauma-focused psychological treatments can be effectively used as first-line therapy for PTSD and mental health comorbidities in people exposed to complex trauma.

- Because phasing of treatment was categorised as a constituent part of multicomponent interventions, there is a case to move beyond binary distinctions of phase-based versus non-phase-based interventions, which has hampered progress in PTSD research.

- Future studies could test the most effective means to deliver patient-centred and multicomponent interventions for people exposed to complex trauma, especially in those with higher levels of mental health comorbidity.

### Introduction

Complex trauma is an increasing threat to global mental health. Complex trauma is defined as exposure to multiple or prolonged traumatic events, typically of an interpersonal nature and from which escape is impossible or difficult. Beyond the prototypical case of childhood sexual abuse, complex-trauma exposure is also common among those who experience intimate partner violence and conflict. Intimate partner violence accounts for 14% of lifetime traumas and is associated with a conditional risk of posttraumatic stress disorder (PTSD) of 11.4%; war-related trauma among military personnel, civilians, and refugees accounts for a further 13.1% of lifetime trauma exposures and is associated with a conditional risk of PTSD of 3.5% [1].

The burden of mental illness among veterans and forcibly displaced people is of critical contemporary relevance. Among United Kingdom veterans, PTSD prevalence has increased from 4% to 6% in the last 10 years and anxiety and depression occur in 31% who held combat roles [2]. UK veterans also report high levels of preservice adversity, and PTSD severity in this population is associated with childhood adversity [3]. Even higher rates of PTSD and mental health comorbidities are reported among forcibly displaced people [4]. A record 70.8 million people were displaced at the end of 2018 and the vast proportion seek refuge and asylum in developing countries with significant implications for health service delivery and budgets [5].

Individual trauma-focused cognitive behavioural therapy (TF-CBT) and eye movement desensitisation and reprocessing (EMDR) therapy are effective for reducing clinician-rated PTSD symptoms [6–8]. Pharmacological treatments are also effective for managing PTSD symptoms but to a lesser degree [9]. However, treatment adherence and recovery rates can be low [10]. There is evidence that complexity of trauma exposure is associated with greater number of different types of comorbid symptoms in addition to PTSD [11, 12], and multiple

comorbidity of symptoms may contribute to poorer outcome. Indeed, high levels of complex psychiatric comorbidities may negatively affect treatment outcomes for people with PTSD [13].

Risk of dropout and reduced treatment efficacy is of particular concern in the presence of complex PTSD (CPTSD), which has recently been recognised by ICD-11 as a new diagnosis. CPTSD includes the core symptoms of PTSD (increased anxiety and emotional arousal, avoidance and numbing, reexperiencing the traumatic event) and additional symptoms associated with disturbances of self-organisation (affective dysregulation, negative self-concept, and interpersonal problems) [14]. A recent meta-analysis of evidence-based therapies for PTSD found that a history of childhood trauma was associated with less beneficial outcomes for all six symptom domains described in CPTSD [15]. These results suggest the importance of exploring the impact of other types of complex-trauma experiences on symptom outcomes. Furthermore, we still do not know which treatment components are most effective and acceptable for people with PTSD following complex-trauma histories.

Because of the narrow analytical focus and limitations of the current evidence base, we conducted a systematic review to identify and integrate all direct and indirect comparisons of psychological and pharmacological treatments versus usual care and active controls in treating mental health problems in people with a history of complex traumatic events. We present post-treatment and short-term effectiveness and acceptability results using pairwise meta-analysis and assessed the relative efficacy of different components of psychological interventions using component network meta-analysis (NMA).

## Methods

The protocol for this study was registered on PROSPERO (CRD42017055523) and can be found at dx.doi.org/10.17504/protocols.io.bdbni2me. We followed the PRISMA extension statement for NMAs (S1 Text) [16].

### Study design and participants

We included randomised controlled trials (RCTs) and non-RCTs of psychological and/or pharmacological interventions for adults with a history of complex traumatic events. Following independent peer review during the development of the protocol, it was agreed with the study steering committee that non-RCTs would be included to capture data on emerging treatments and treatments tested in more pragmatic settings. Complex traumatic events were defined as extreme and prolonged or repetitive in nature and experienced as extremely threatening or horrific and difficult or impossible to escape from [17]. Inclusion was based on the type of exposure rather than the ICD-11 diagnostic category of CPTSD. Candidate exposures included (but were not limited to) childhood physical and/or sexual abuse, domestic violence, forcible displacement, torture, ongoing armed conflict and combat, and human trafficking.

### Interventions and comparators

First- or second-line psychological therapies aimed at improving PTSD symptoms and mental health comorbidities delivered either to individuals or in a group were included. As per our protocol and in keeping with the classification used by the National Institute for Health and Care Excellence (NICE) [6], interventions considered were (1) TF-CBT that included one or more of exposure, cognitive therapy, stress management; (2) EMDR; (3) other psychological treatments used to treat trauma survivors but that use predominately non-CBT techniques such as supportive therapy and nondirective counselling, interpersonal psychotherapy (IPT), hypnotherapy, mindfulness- and compassion-focused therapies, acceptance and commitment

therapies, accelerated resolution, and sensorimotor therapies. We also included the following pharmacological interventions: antidepressants (selective serotonin reuptake inhibitors [SSRIs]; tricyclics and monoamine oxidase inhibitors), antipsychotics (quetiapine, aripiprazole, risperidone, olanzapine), hypnotics and anxiolytics (Z-drugs; benzodiazepines; promethazine), alpha blocker and antihypertensive (prazosin), and anticonvulsants (lamotrigine, topiramate, valproate).

Comparators for psychological interventions were waitlist; treatment as usual (defined as nonexperimental active treatments that conform to best and/or clinical guideline–recommended care ordinarily made available to patients); no intervention; symptom monitoring; repeated assessment or other minimal attention control group akin to psychological placebo; and alternative psychological treatment. Comparators for pharmacological interventions were placebo; other medication; no intervention; and psychological therapy.

Comparisons of two or more active interventions were included. Differences in comparators were taken into account during data summary and analyses. NMAs were conducted to provide comparisons of all interventions within a connected network (including comparisons of active interventions not originally evaluated in included trials).

## Outcomes

The primary outcome was reduction in severity of PTSD symptoms as measured using a validated and standardised clinician-rated scale. Secondary outcomes were reductions in symptoms of disturbances of self-organisation (affect dysregulation; negative self-concept; disturbances in relationships); reduction in symptoms of depression and anxiety, dissociation, functional somatic syndromes; acceptability (attrition); adverse events and harms from trial data (e.g., worsening of traumatic stress symptoms); suicidal ideation, attempts, and completion; and quality of life measured by validated clinician-rated scales. Study outcomes were measured at posttreatment and/or at the follow-up point defined by the study.

## Search strategy and selection criteria

Literature searches were initially conducted in April 2017 in these databases: CINAHL, Cochrane Central Register of Controlled Trials (CENTRAL), Embase, International Pharmaceutical Abstracts, MEDLINE, Published International Literature on Traumatic Stress (PILOTS), PsycINFO, and Science Citation Index. The search results for each database were downloaded, imported into EndNote bibliographic software and deduplicated. A full update search was conducted in August 2018. Finally, update searches using the MEDLINE and PsycINFO databases were carried out in October 2019. Details of search dates, database interfaces, and the full search strategies used are available from the corresponding author. We did not restrict on language and translated studies where feasible, but we did not search Chinese databases or translate this language. A sample MEDLINE search is shown in S2 Text.

Studies were eligible if they met these criteria: (1) peer-reviewed original articles; (2) RCTs and non-RCTs; (3) measured either the primary or one of the candidate secondary outcomes. The exclusion criteria were (1) reviews/non-original data; (2) dissertations or conference presentations; (3) complementary and alternative therapeutic interventions that were not underpinned by a recognisable psychological focus (i.e., yoga; dance, music, art). To ensure that the inclusion criteria were consistently applied, a 10% sample of records was first double-screened based on title and abstract by pairs of researchers. Consensus meetings with the rest of the research team were held at regular intervals to resolve unclear decisions at the title and abstract screening phase. Full text records were similarly screened with consensus meetings used to resolve disagreements.

## Data extraction

Data extraction was piloted on a small sample of studies by three researchers independently. Both RCTs and non–randomised controlled studies were extracted using the same template and managed in separate Excel spreadsheets. After consensus checking, included records were split between three reviewers to singly extract, owing to the volume of evidence. Uncertainties were resolved by consultation between reviewers tasked with data extraction or by deferring to the wider review team. Extracted data across domains related to study and participant characteristics and outcomes were compiled in a spreadsheet. Where presented, intention-to-treat data were extracted instead of complete cases.

Where an included study was published across multiple manuscripts, we used the primary publication as the main source of information. New and follow-up data were taken from subsequent publications, but the unit of allocation remained the study rather than numbers of publications.

## Risk of bias

Risk of bias for RCTs was assessed with the Cochrane Risk of Bias tool [18]. This tool assessed each study against domains known to be associated with bias in RCTs: selection, performance, detection, attrition, reporting, and other bias (which was applied based on the specific context). Each study was assessed as being at either 'low', 'unclear', or 'high' risk of bias across each of these domains. Attrition bias was used as an independent variable in the sensitivity analysis; this domain was checked by a further reviewer after all the original appraisals had been made. Overall, RCTs were classified as having a low risk of bias if none of the domains were rated as high risk of bias and three or less domains were rated as unclear risk, and RCTs were classified having a moderate risk of bias if one domain was rated as high risk of bias or if no domain was rated as high risk of bias but four or more domains were rated as unclear risk. All other cases were assumed to be at high risk of bias [19].

Studies of non-RCTs were assessed for risk of bias using a modified version of the NICE (2012) quality appraisal checklist [20]. This checklist was originally developed based on the Graphic Appraisal Tool for Epidemiological studies (GATE) tool and includes domains of population bias, allocation, outcomes, and analyses, as well as summary judgements for internal and external validity [21].

## Statistical analysis

Random-effects pairwise meta-analyses were conducted using Stata 15 [22]. Control conditions were grouped into two categories: control (which included waitlist, usual care, no treatment, or other control with no or minimal therapeutic input) and active control (attention controls or treatment as usual with non–systematic psychological intervention input). Where multiple intervention groups were included in the study, we analysed the data in the following way: (1) if one of the groups did not meet criteria for our review, we did not combine across groups but used data from the group that met our review criteria; (2) where studies included two intervention groups that met criteria for the same intervention classification, we combined them together. For example, if a study included a prolonged exposure group and a cognitive processing therapy group, we combined them together into one group for the TF-CBT analyses.

Most outcomes were continuous. Where all studies used the same scale, we calculated mean differences (MDs) and their 95% confidence interval (CI). Where studies used different scales to measure a particular outcome we calculated standardised MDs (SMDs) and their 95% CI. In keeping with established cutoffs of effect in behavioural medicine, SMDs of 0.56–1.2 were

categorised as large, effect sizes of 0.33–0.55 as moderate, and effect sizes of ≤0.32 as small [23]. For dichotomous outcomes, such as attrition, we calculated odds ratios (ORs) and their 95% CI. Heterogeneity assessment was based on visual inspection of forest plots and the $I^2$ statistic [24]. A Q-value (approximating $X^2$ distribution) of $p < 0.1$ indicated statistically significant heterogeneity. Statistical heterogeneity was explored using subgroup analyses and components NMAs.

Given the substantial and inherent heterogeneity expected from our broad research questions, we conducted a range of subgroup analyses. Firstly, we conducted meta-analyses including all psychological interventions vs inactive controls or active controls in all populations. Secondly, we subgrouped these meta-analyses of all psychological interventions into the following populations based on descriptions in the study and through discussion with clinical experts: veterans, people who had experienced childhood sexual abuse, refugees, people who had experienced domestic violence, and war-affected civilians. Thirdly, we subgrouped the data according to intervention categories commonly reported in the literature based on reporting from the original papers and discussion with clinical experts: TF-CBT, EMDR, non-trauma-focused CBT, mindfulness, dialectical behaviour therapy (DBT), and IPT.

We sought to further explore the impact of different combinations of psychological intervention components using NMAs. We used a Bayesian approach, as this allows greater flexibility in fitting more complex models and aids exploration of heterogeneity. Given the greater complexity of the NMA models, we simplified the analyses by focusing on MDs for the Clinician-Administered PTSD Scale in all populations for this outcome.

We fitted models using WinBUGS 1.4.3 based on the components NMAs approach proposed by Welton and colleagues [25] and an adaptation of the WinBUGS code reported by Freeman and colleagues [26]. The advantages of this approach is that all intervention components can be included in the meta-analyses as long as they form a connected network. An important assumption of the NMA is consistency between direct (i.e., where trials have specifically compared two or more interventions) and indirect (i.e., data derived from the network where trials have not directly compared interventions) evidence. To assess the validity of this assumption, we examined participant and study characteristics and sought input from topic experts. Based on this assessment, we judged the data similar enough to combine in the NMA. However, as is common in most NMAs, there was insufficient data to statistically test this assumption.

All models used a normal likelihood for continuous outcomes and vague priors for treatment effect and between-trial SD. Convergence was assessed based on visual assessment of trace plots, the Brooks-Gelman-Rubin statistic, and autocorrelation plots using three Markov chain Monte Carlo chains. All models were judged to have reached convergence after 50,000 iterations. These iterations were then discarded and all results were based on a further 50,000 iterations.

Goodness of fit to the observed data was assessed using total residual deviance and the deviance information criterion (DIC). Total residual deviance approximately equal to the number of data points was considered to indicate acceptable fit [27]. Greater than five points on the DIC was considered a substantial difference in goodness of fit between models [28].

We compared four models: (1) Model 1 included the intervention categories used in the pairwise meta-analyses (TF-CBT, EMDR, non-trauma-focused CBT, mindfulness, and IPT) compared with either control or active control. (2) Model 2 included all intervention components included in the intervention categories from model 1 (support, psychoeducation, relaxation, cognitive restructuring, in vivo exposure, imaginal exposure, virtual reality exposure, mindfulness, phase-based). In addition to these, it was also assumed that all active treatments and attention controls included a placebo component. We also took into account the effect of

control group (waitlist versus active control). Each component had a separate effect and assumed the total effect of the intervention was a sum of these separate effects. (3) Model 3 included all intervention components in model 2 plus all available pairs of components. Ten pairs of intervention components were reported in two or more included studies—support + psychoeducation, psychoeducation + relaxation, psychoeducation + cognitive restructuring, psychoeducation + imaginal exposure, relaxation + mindfulness, relaxation + cognitive restructuring, relaxation + imaginal exposure, mindfulness + cognitive restructuring, cognitive restructuring + in vivo exposure, cognitive restructuring + imaginal exposure—and were therefore included in the analyses. This model allowed for interactions between pairs of interventions above or below what would be expected from the sum of their components. (4) Model 4 included all possible combinations of intervention components.

For the attrition outcome, we were concerned that any differences between interventions and control may be confounded by study design characteristics. Therefore, we conducted sensitivity analyses on attrition outcomes, including only studies with low risk of attrition bias, and compared these findings with all included studies.

## Results

### Characteristics of the included studies

A total of 11,845 nonduplicate references were identified by the search (last update 25 October 2019), and 518 full text articles were assessed for eligibility (Fig 1). We included 116 studies (115 papers) in the systematic review. Of these, 50 were conducted in hospital settings [29–78], 24 were delivered in a community setting [79–102], seven were delivered in military clinics for veterans or active military personnel [103–109], five were conducted in refugee camps [110–114], four used remote delivery via web-based or telephone platforms [115–118], four were conducted in specialist trauma clinics [119–122], two were delivered in home settings [123, 124], and two were delivered in primary care clinics [125, 126]; clinical setting was not reported in 17 studies [127–143].

Ninety-four (*n* = 6,158 participants) RCTs were included in meta-analyses across the primary and secondary outcomes. Eighteen RCTs (*n* = 933 participants) of psychological interventions that measured the primary outcome with CAPS were included in the NMA [29, 36, 39, 44, 59, 68, 84, 88, 91–93, 100, 106, 107, 109, 116, 120, 123]. The complex-trauma subgroups of the included studies were categorised as follows: post–combat deployment veterans (55 studies) [32–35, 37, 39–41, 43–48, 50–54, 56, 58, 60–63, 66–71, 73, 74, 76, 77, 82, 90, 100, 103, 104, 106–108, 115, 116, 121, 123, 124, 127, 128, 132, 133, 136, 143]; war-related (16 studies; 15 papers) [30, 79, 80, 86, 96, 101, 102, 109, 117, 118, 122, 125, 126, 134, 139]; childhood sexual abuse (17 studies) [36, 38, 49, 55, 57, 59, 72, 84, 91, 95, 97, 98, 129, 135, 141, 142]; refugees (19 studies) [29, 64, 65, 75, 81, 83, 87–89, 94, 99, 110–114, 119, 120, 140]; domestic violence (5 studies) [31, 92, 93, 131, 137]; and mixed presentation (4 studies) [78, 85, 105, 130]. The mean age of participants in the included RCTs was 42.6 ± 9.3 years, and 42% were male (S1 Table).

Across the 51 (*n* = 4,018 participants) RCTs of psychological interventions included in the meta-analyses of the primary outcome, there were 27 comparisons of TF-CBT, nine comparisons of EMDR, two comparisons of IPT, three comparisons of mindfulness, three comparisons of non-trauma-focused CBT, and seven comparisons of DBT. TF-CBT was delivered over a mean of 10.3 weeks with an average of 1.2 sessions a week, lasting on average 59.4 minutes. Non-trauma-focused CBT was delivered over a mean of 12 weeks with an average of 1.5 sessions a week for an average of 68.6 minutes. The duration of EMDR was shorter, delivered over a mean of 5.2 weeks, with an average of 1.1 sessions a week for an average of 61 minutes each. Mindfulness was delivered over a mean of 6.6 weeks, with an average of 1.1 sessions a

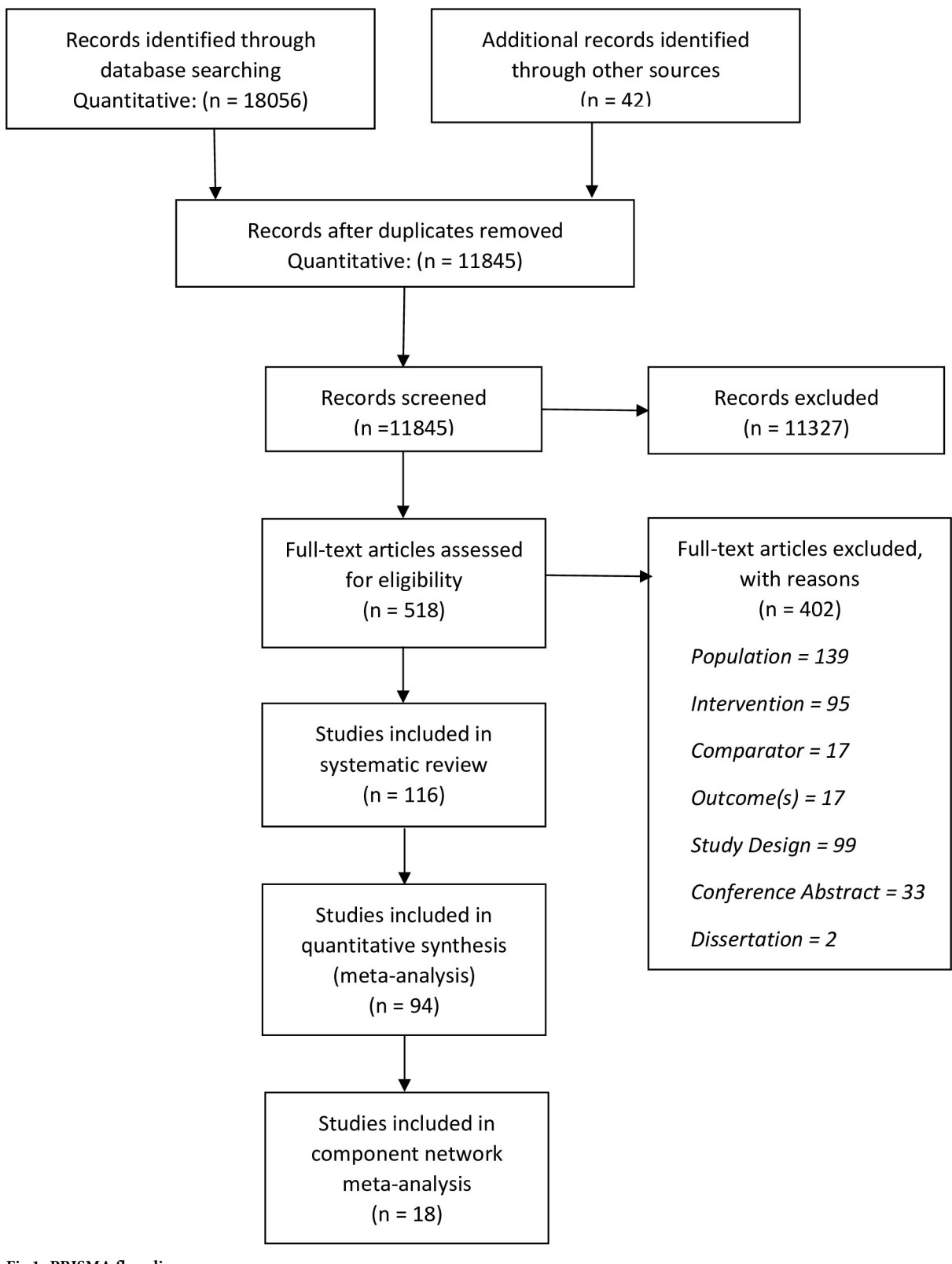

**Fig 1. PRISMA flow diagram.**

week lasting an average of 121.6 minutes per session. There was insufficient data to report mean duration, frequency, and length of sessions for IPT and DBT.

Sixteen (*n* = 1,233 participants) of 19 RCTs contributed data to meta-analyses of pharmacological interventions versus placebo. These studies included six comparisons of antidepressants (of these, four comparisons were of SSRIs), five comparisons of antipsychotics, two comparisons of anticonvulsants, and three comparisons of prazosin. Of those studies that compared SSRIs with a placebo control, there was only sufficient data from trials that tested sertraline and paroxetine to report mean duration, frequency, and dosing. Sertraline was prescribed for a mean of 9.5 weeks, to be taken daily, with a mean dose of 50 mg. Paroxetine was prescribed for a mean of 8.6 weeks, to be taken daily, with a mean dose of 30 mg.

Nine non-RCTs were included, and of these, six reported data for the primary outcome [52, 57, 66, 95, 96, 132–134, 138]. The mean age of participants in the non-RCTs was 40.6 ± 9.4 years, and 47% were male. Effect sizes were calculated for four of these studies (representing five interventions), as they used inactive control comparators. All comparisons were of TF-CBT.

Of the 22 RCTs not included in the meta-analyses, five studies compared psychological interventions in veterans. Of these, two studies compared TF-CBT with present-centred therapy and one study compared mindfulness with present-centred therapy [67, 108]. Additionally, one study compared TF-CBT with exposure alone and another study did not include extractable data [103]. Two RCTs were identified that compared combined psychological and pharmacological interventions but included different classes of drugs. Of these, one study was in veterans and compared phenelzine and psychotherapy with imipramine and psychotherapy and with psychotherapy alone [90]. A further study was in a mixed population and compared tianeptine and group therapy with fluoxetine and group therapy [130]. Three RCTs in veterans that compared pharmacological interventions were not included in the meta-analyses. Of these, one study compared rivastigmine-augmented therapy with placebo, but there were no other comparable interventions to combine these data with [127]. Two other studies were head-to-head comparisons of paroxetine with amitriptyline [35] and of mirtazapine with sertraline [37].

Three RCTs in refugees were not meta-analysed. One study compared TF-CBT, supportive counselling, and psychoeducation and did not include a comparison with a control group [112]. Another study compared TF-CBT with an exposure-only intervention [65], and one comparison of TF-CBT with treatment as usual did not include extractable data [75]. Among RCTs that assessed anxiety in refugees, three studies compared combined psychological and pharmacological interventions, but no meta-analyses were possible [64, 81, 99]. Additionally, one RCT in refugees compared paroxetine with sertraline, but this was the only study in this subgroup that used this comparison, and no meta-analysis was possible [140].

In RCTs among war-affected populations, one study did not report outcomes that were similar enough with other studies [30], and another study used a head-to-head design that compared TF-CBT with psychoeducation [79].

Four RCTs in populations with a history of childhood sexual abuse were not included in meta-analyses. One study attempted to deconstruct how skills training drove the effectiveness and interacted with counselling and exposure, respectively, and did not offer opportunities to formally compare outcomes with an inactive or active control group [84]. A head-to-head design was used by one study to compare analytic group psychotherapy with systemic group psychotherapy [55], whereas another study combined data from TF-CBT and present-centred therapy, making it difficult to extract relevant data [38]. A further study that compared TF-CBT with a minimal attention control group did not include data that could be compared with other studies [135].

### Risk of bias assessment

Forty, 25, and 42 RCTs were categorised as being of low, moderate, and high risk of bias, respectively. For RCTs, the risk of bias from random sequence generation was low in 35 (32%) studies and low for allocation concealment in 12 (11%). Two, four, and three non-RCTs were categorised as being of low, moderate, and high risk of bias. For non-RCTs, risk of bias associated with selection bias was low in only two studies (11%). A breakdown of risk of bias by individual domains for RCTs is shown in S2 Table and for non-RCTs in S3 Table.

### Acceptability

The acceptability sensitivity analysis showed that participants across all populations allocated to psychological interventions in studies judged to be at low risk of attrition bias were still less likely to drop out compared with controls (OR = 0.39; 0.21–0.73) than in all studies (OR = 0.56; 0.40–0.80).

### Primary outcome: PTSD symptoms

**Effectiveness at posttreatment.** The pairwise meta-analysis results for primary and secondary outcomes across all populations at posttreatment and follow-up versus control are shown in S4 Table. Across 46 trials in all populations, psychological treatments were effective at posttreatment in reducing PTSD symptoms in people with a history of complex traumatic events (Fig 2). Across all populations, TF-CBT, IPT, and EMDR were associated with large

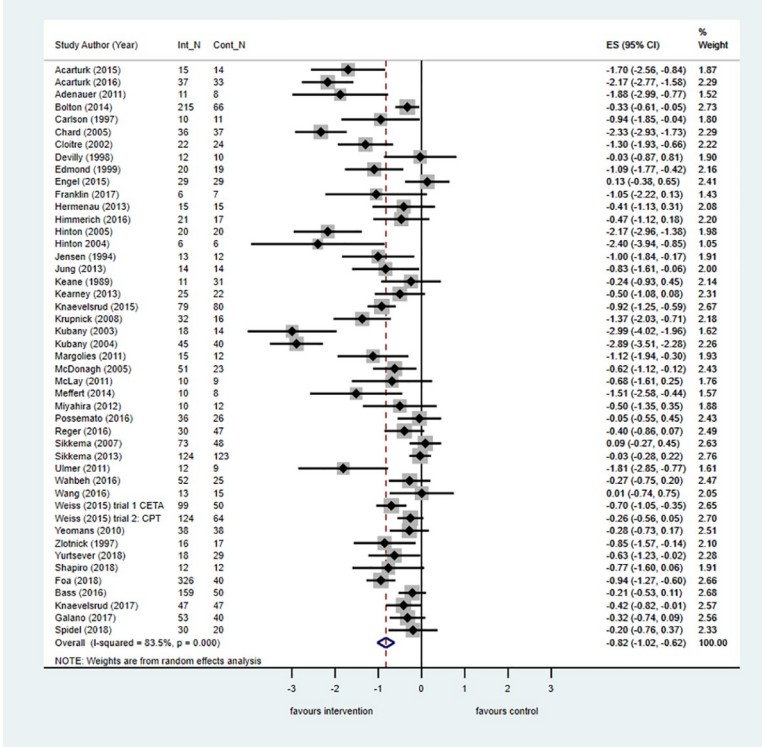

**Fig 2. Any psychological treatment for PTSD symptoms versus control at posttreatment across all populations.** The size of the grey box reflects how much weight each study received in the meta-analysis (i.e., the larger the box, the more this study contributed to the pooled effect represented by the blue diamond). Black bars represent the 95% CI for the effect size in each study. CI, confidence interval; Cont_N, number in control group; ES, effect size; Int_N, number in intervention group; PTSD, posttraumatic stress disorder.

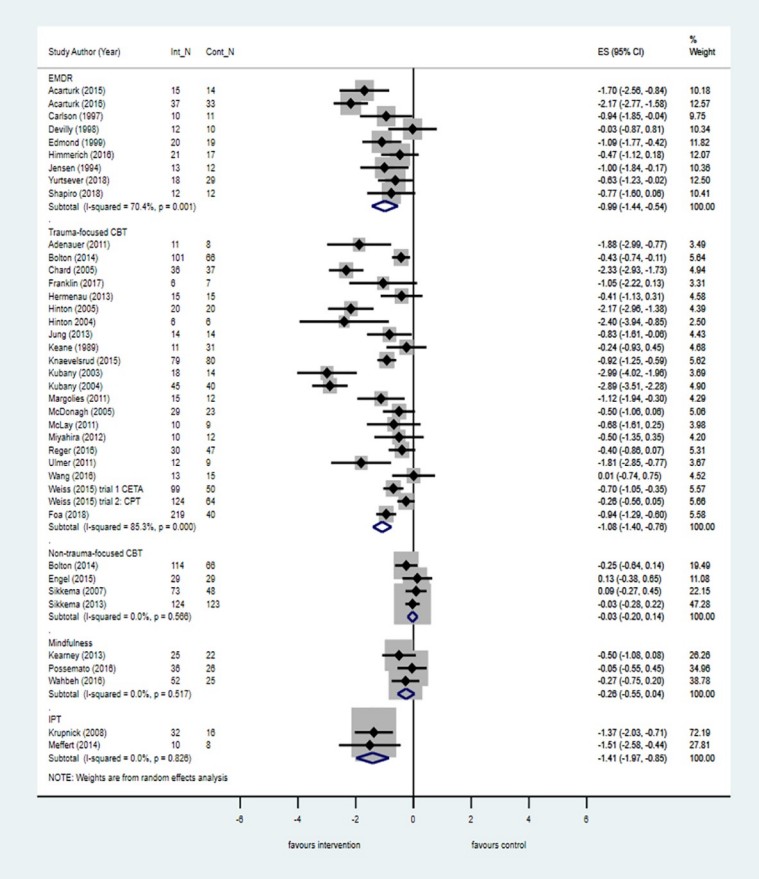

**Fig 3. Psychological treatments for PTSD symptoms by intervention category versus control at posttreatment across all populations.** The size of the grey box reflects how much weight each study received in the meta-analysis (i.e., the larger the box the more this study contributed to the pooled effect represented by the blue diamond). Black bars represent the 95% CI for the ES in each study. CBT, cognitive behavioural therapy; CI, confidence interval; Cont_N, number in control group; EMDR, eye movement desensitisation and reprocessing therapy, ES, effect size; Int_N, number in intervention group; IPT, interpersonal therapy; PTSD, posttraumatic stress disorder.

treatment effects in favour of the interventions at posttreatment when compared with control (Fig 3). The 95% CIs for IPT were large, suggesting substantial imprecision. Smaller but still significant effects were observed at posttreatment when TF-CBT was compared with an active control (k = 3; $n$ = 447; SMD = −0.30; −0.50 to −0.10; $I^2$ = 13.2%, $p$ = 0.32). There was also evidence from six trials that phase-based interventions that included components to improve daily functioning as well as trauma-focused therapy were effective at reducing PTSD symptoms at posttreatment compared with control. Treatment effects associated with non-trauma-focused interventions were small and not significant.

Eight trials compared pharmacological interventions with placebo for reducing PTSD symptoms. Overall, antipsychotic medicine (k = 5; $n$ = 364; SMD = −0.45; −0.85 to −0.05; $I^2$ = 51.2%, $p$ = 0.085) (Fig 4) and prazosin (k = 3; $n$ = 110; SMD = −0.52; −1.03 to −0.02; $I^2$ = 41.4%, $p$ = 0.182) (Fig 5) were effective in reducing PTSD symptoms.

**Effectiveness at 6-month follow-up.** All psychological treatments were effective compared with control at 6-month follow-up (k = 10; $n$ = 738; SMD = −0.45; −0.82 to −0.08; $I^2$ = 79.4%; $p$ < .001). There was further evidence from four trials that TF-CBT conferred the most

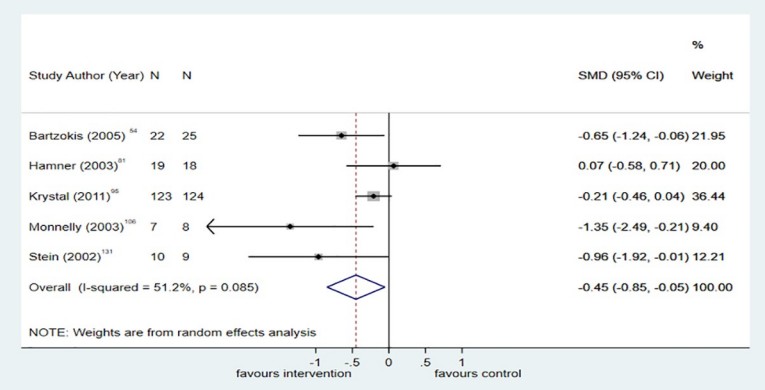

**Fig 4. Antipsychotics versus placebo for PTSD symptoms at posttreatment.** The size of the grey box reflects how much weight each study received in the meta-analysis (i.e., the larger the box, the more this study contributed to the pooled effect represented by the blue diamond). Black bars represent the 95% CI for the effect size in each study. CI, confidence interval; PTSD, posttraumatic stress disorder; SMD, standardised mean difference.

benefit, with large treatment effects reported at 6-month follow-up (k = 4; $n$ = 206; SMD = −0.64; −1.10 to −0.18; $I^2$ = 44.9%; $p$ = 0.14).

## Subgroup analyses

The pairwise meta-analyses results for the primary outcome by subgroup are presented in S5 Table. It was not possible to conduct meta-analyses for pharmacological interventions by population, as all but one of these studies were conducted in veterans.

**Veterans.** Among veterans, evidence from 15 trials showed that psychological interventions compared with control were effective at posttreatment for reducing PTSD symptoms, but the size of the treatment effect was smaller than in the pooled analysis across all populations. Additionally, unlike the pooled analysis across all populations, these positive effects were not maintained at 6-month follow-up. However, when compared with an active control in six trials, psychological interventions were associated with a moderate and significant effect size at posttreatment (k = 6; $n$ = 260; SMD = −0.40; −0.77 to −0.02; $I^2$ = 48.7%, $p$ = 0.08). Results by intervention category are shown in Fig 6. In seven trials and four trials, respectively, TF-CBT

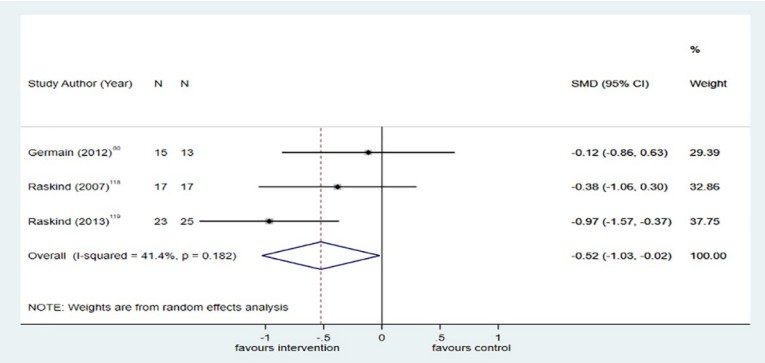

**Fig 5. Prazosin versus placebo for PTSD symptoms at posttreatment.** The size of the grey box reflects how much weight each study received in the meta-analysis (i.e., the larger the box, the more this study contributed to the pooled effect represented by the blue diamond). Black bars represent the 95% CI for the effect size in each study. CI, confidence interval; PTSD, posttraumatic stress disorder; SMD, standardised mean difference.

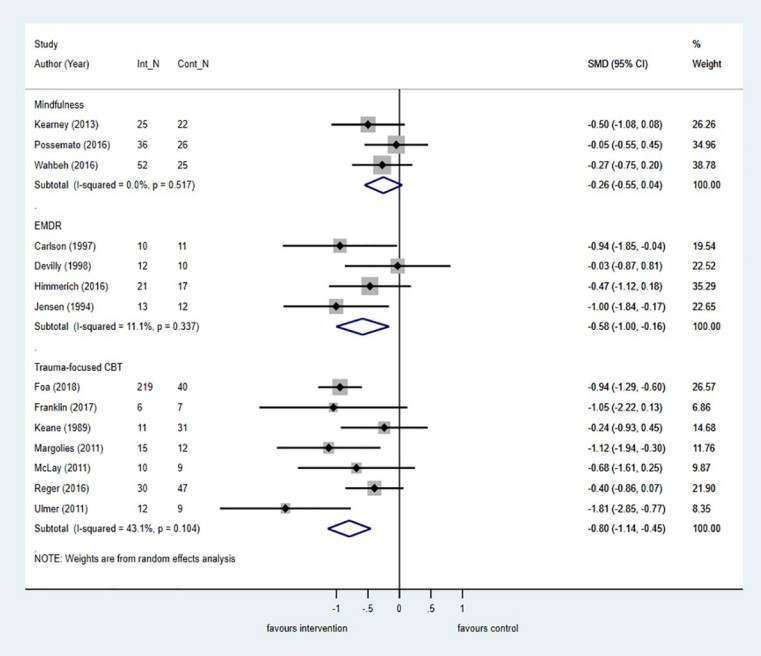

**Fig 6. Psychological treatments for PTSD symptoms by intervention category versus control at posttreatment in veterans.** The size of the grey box reflects how much weight each study received in the meta-analysis (i.e., the larger the box, the more this study contributed to the pooled effect represented by the blue diamond). Black bars represent the 95% CI for the effect size in each study. CBT, cognitive behavioural therapy; CI, confidence interval; Cont_N, number in control group; EMDR, eye movement desensitisation and reprocessing therapy; Int_N, number in intervention group; PTSD, posttraumatic stress disorder; SMD, standardised mean difference.

and EMDR were associated with the largest treatment effect at posttreatment compared with control, but the effect size was reduced by one-third when compared with the pooled analysis across all populations. Treatment effects associated with mindfulness favoured the intervention at posttreatment and 6-month follow-up compared with control, but the difference was not significant in either comparison.

**Refugees.** Psychological interventions are effective for reducing PTSD symptoms in refugee populations in seven trials at posttreatment and in three trials at 6-month follow-up compared with control. Evidence from two trials showed that TF-CBT conferred the most benefit at posttreatment compared with control, but the large effects were not maintained in two trials at 6-month follow-up. EMDR was also associated with large and significant treatment effects in three trials at posttreatment when compared with control (Fig 7).

Non-trauma-focused CBT was investigated in one non-RCT in a refugee population and showed a large and significant effect favouring group intervention for reducing PTSD symptoms (k = 1; $n$ = 43; SMD = −2.54, −3.21 to −1.88).

**Childhood sexual abuse.** Across 10 trials, psychological interventions were effective in reducing PTSD symptoms in childhood sexual abuse populations when compared with control at posttreatment, but the difference was not significant in three trials that evaluated outcomes at 6-month follow-up. When broken down by treatment type, only TF-CBT was associated with positive and significant effects in three trials that compared outcomes at posttreatment with control (k = 3; $n$ = 153; SMD = −1.22; −2.40 to −0.05; $I^2$ = 90.3%, $p$ = 0.000), but the wide 95% CIs suggest significant imprecision in this estimate.

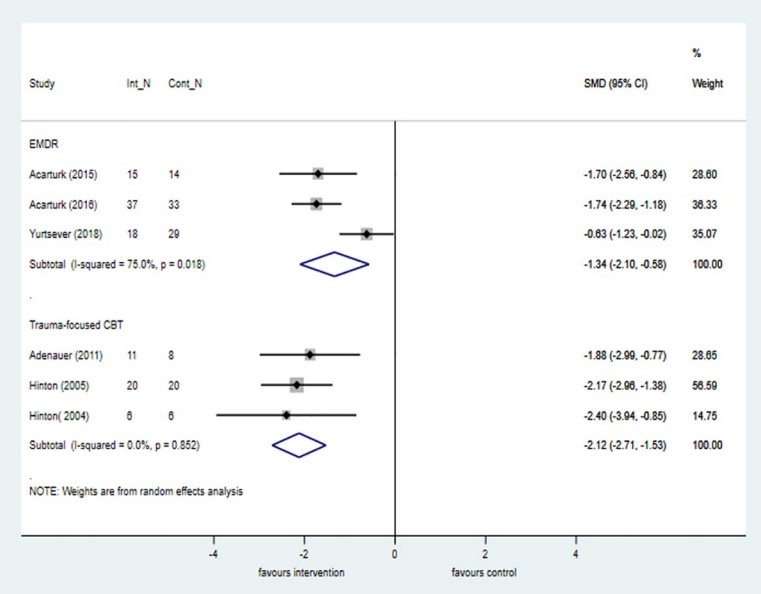

**Fig 7. Psychological treatments for PTSD symptoms by intervention category versus control at posttreatment in refugee populations.** The size of the grey box reflects how much weight each study received in the meta-analysis (i.e., the larger the box, the more this study contributed to the pooled effect represented by the blue diamond). Black bars represent the 95% CI for the effect size in each study. CBT, cognitive behavioural therapy; CI, confidence interval; Cont_N, number in control group; EMDR, eye movement desensitisation and reprocessing therapy; Int_N, number in intervention group; PTSD, posttraumatic stress disorder; SMD, standardised mean difference.

Evidence from non-RCTs revealed a similar pattern. One study investigated 'victim to survivor' group TF-CBT therapy, and treatment effects were large and favoured the intervention at posttreatment (k = 1; $n$ = 45; SMD = −1.01; −1.53 to −0.48). Another study examined a multicomponent trauma-focused intervention delivered in a group format; a small reduction in PTSD symptoms was found, but this was not significant (k = 1; $n$ = 63; SMD = −0.18; −0.62 to 0.26).

**War-related.** Evidence from six trials shows that TF-CBT is effective compared with control at posttreatment in reducing PTSD symptoms in populations affected by war. The size of the treatment effect was approximately half that observed in the comparable analysis that pooled data across all populations (Fig 8). Trauma-focused approaches were investigated in one non-RCT, which showed large treatment effects in favour of the intervention at posttreatment compared with control (k = 1; $n$ = 115; SMD = −1.22; −1.75 to −0.69).

**Domestic violence.** TF-CBT was the most effective intervention for reducing PTSD symptoms in people exposed to domestic violence, with large and significant treatment effects observed across two trials (k = 2; $n$ = 117; SMD = −2.92; −3.45 to −2.39; $I^2$ = 0%, $p$ = 0.970).

## Secondary outcomes

The pairwise meta-analyses results for the secondary outcomes by subgroup are presented in S5 Table. Only outcomes that were meta-analysed are reported.

**Disturbances of self-organisation symptoms.** Evidence from seven trials showed that treatment effects favoured psychological interventions for reducing symptoms of emotional dysregulation compared with control at posttreatment and 6-month follow-up, but the differences were not significant. Evidence from two trials showed that phase-based interventions were associated with large treatment effects in favour of reducing interpersonal problems, but

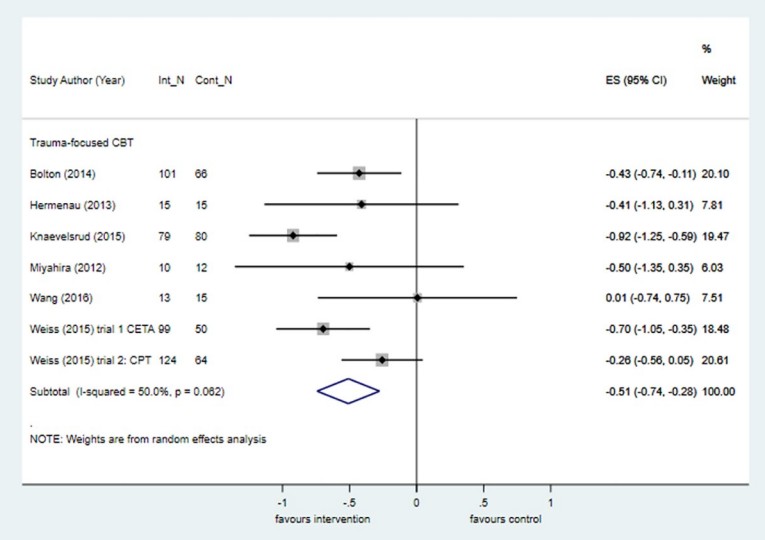

**Fig 8. Trauma-focused CBT for PTSD symptoms versus control at posttreatment in war-affected populations.**
The size of the grey box reflects how much weight each study received in the meta-analysis (i.e., the larger the box, the more this study contributed to the pooled effect represented by the blue diamond). Black bars represent the 95% CI for the ES in each study. CBT, cognitive behavioural therapy; CI, confidence interval; Cont_N, number in control group; ES, effect size; Int_N, number in intervention group; PTSD, posttraumatic stress disorder.

the difference was not significant. Across five trials, negative self-concept was significantly improved by any psychological intervention at posttreatment compared with control (k = 5; $n$ = 215; SMD = 1.81; 0.73–2.89; $I^2$ = 90%, $p$ = 0.000). TF-CBT was associated with large treatment effects at posttreatment compared with control in favour of improving negative self-concept (k = 3; $n$ = 145; SMD = 2.22; 0.75–3.70; $I^2$ = 90.4%, $p$ = 0.000), but the wide 95% CIs suggest this estimate is potentially imprecise. No studies evaluated the effect of pharmacological therapies for these outcomes.

**Depression.** Across all populations, evidence from 31 and 6 trials respectively showed psychological interventions are effective for reducing depressive symptoms at posttreatment and 6-month follow-up when compared with control. Smaller positive effects were seen across five trials that compared psychological interventions at posttreatment with an active control, but the difference was not significant (k = 5; $n$ = 473; SMD = −0.38; −0.76 to 0.01; $I^2$ = 70.5%, $p$ = 0.009). TF-CBT was associated with the most consistently large and significant treatment effects in favour of reducing depressive symptoms at posttreatment and 6-month follow-up compared with control; in two trials, TF-CBT was also effective at posttreatment when compared with an active control (k = 2; $n$ = 346; SMD = −0.60; −1.06 to −0.14; $I^2$ = 77.7%, $p$ = 0.03). In seven trials, EMDR was similarly associated with large and significant treatment effects for reducing depressive symptoms across all populations when compared with control at posttreatment; smaller effects were observed in two trials that compared EMDR with an active control but the difference was not significant (k = 2; $n$ = 72; SMD = −0.32; −1.23 to 0.59; $I^2$ = 47.8%, $p$ = 0.17). Large and significant effects were observed in two trials that compared IPT with control at posttreatment across all populations. Similarly, evidence from four trials showed that phase-based interventions were associated with large and significant treatment effects at posttreatment when compared with control. Mindfulness was another non-trauma-based intervention that proved moderately effective for reducing depressive symptoms across three trials at posttreatment and two trials at 6-month follow-up.

When broken down by trauma exposure, evidence from three trials showed that TF-CBT is the most effective trauma-focused intervention for reducing depressive symptoms among veterans, war-affected populations, childhood sexual abuse, refugees, and domestic violence. The size of the treatment effect among veterans and war-affected populations was attenuated compared with the pooled analysis across all populations at posttreatment compared with control. Mindfulness was shown to be moderately effective among veterans at posttreatment compared with control, but this difference was not significant at 6-month follow-up.

**Anxiety.** Across all populations psychological interventions were shown to be effective in 15 trials for reducing anxiety symptoms at posttreatment compared with control; two trials contributed evidence that showed that psychological interventions were moderately effective when compared with an active control (k = 2; $n$ = 346; SMD = −0.44; −0.73 to −0.15; $I^2$ = 46.4%, $p$ = 0.17). For all trauma types, large and significant treatment effects were observed when TF-CBT and EMDR were compared with control in eight and four trials, respectively. Among veterans, TF-CBT (k = 3; $n$ = 112; SMD = −1.02; −1.72 to −0.32; $I^2$ = 51%; $p$ = 0.130) and EMDR (k = 2; $n$ = 44; SMD = −0.91; −2.28 to −0.47; $I^2$ = 77.7%; $p$ = 0.034) were associated with the largest treatment effects for reducing anxiety symptoms when compared with control at posttreatment. TF-CBT was also the most effective intervention for reducing anxiety symptoms among war-affected populations when compared with control at posttreatment in six trials.

**Quality of life.** For all trauma types, small but nonsignificant improvements in quality of life were observed in six trials that compared all different psychological interventions (k = 6; $n$ = 406; SMD = −0.33, 95% CI −0.01 to 0.66; $I^2$ = 57.3%; $p$ = 0.021) and four trials that compared TF-CBT with control at posttreatment (k = 4; $n$ = 260; SMD = 0.23, 95% CI −0.33 to 0.79; $I^2$ = 73.9%; $p$ = 0.009).

**Sleep quality.** Across all trauma types, sleep quality was significantly improved in analyses of three trials of psychological interventions and two trials of TF-CBT at posttreatment compared with control. Prazosin was the only pharmacological intervention with sufficient data to conduct meta-analysis. In three trials, prazosin was effective compared with placebo for improving sleep quality (k = 3; $n$ = 109; SMD = −0.73; −1.12 to −0.34; $I^2$ = 0%, $p$ = 0.486).

**Positive and negative affect.** Evidence from three trials showed that antipsychotic medication (all risperidone) was not effective at posttreatment in improving negative (k = 2; $n$ = 284; SMD = 0.54; 95% CI −0.14 to 1.22; $I^2$ = 0%; $p$ = 0.66) and positive affect (k = 3; $n$ = 329; SMD = 1.75, 95% CI −4.05 to 0.54; $I^2$ = 76.9%; $p$ = 0.01) or general psychopathology symptoms (k = 2; $n$ = 284; SMD = 0.04, 95% CI −2.08 to 2.16; $I^2$ = 0%; $p$ = 0.43) in people with complex trauma.

## Component NMA

We further explored the treatment effects of different psychological components of the included composite complex interventions by using component NMA. Model 2 had the lowest DIC (262.7, SD = 8.6). However, model 3 had a comparable DIC and a substantially lower between-study SD (DIC = 265.5, SD = 6.0), suggesting heterogeneity was better accounted for. The total residual deviance was also lower in model 3, suggesting a better fit between the model and data. Given that the difference in DIC was less than three points, we selected model 3 for further analyses.

Fig 9 shows the network plot of combinations of treatment components for the primary outcome across the 18 studies included in the network [29, 36, 39, 44, 59, 68, 84, 88, 91–93, 100, 106, 107, 109, 116, 120, 123]. MDs for the primary outcome by intervention component are shown in S6 Table. Interventions that took a multicomponent approach were more effective than those that did not for reducing PTSD symptoms (k = 17; $n$ = 1,077; MD = −37.95; −60.84 to −15.16). All these studies included cognitive restructuring and imaginal exposure.

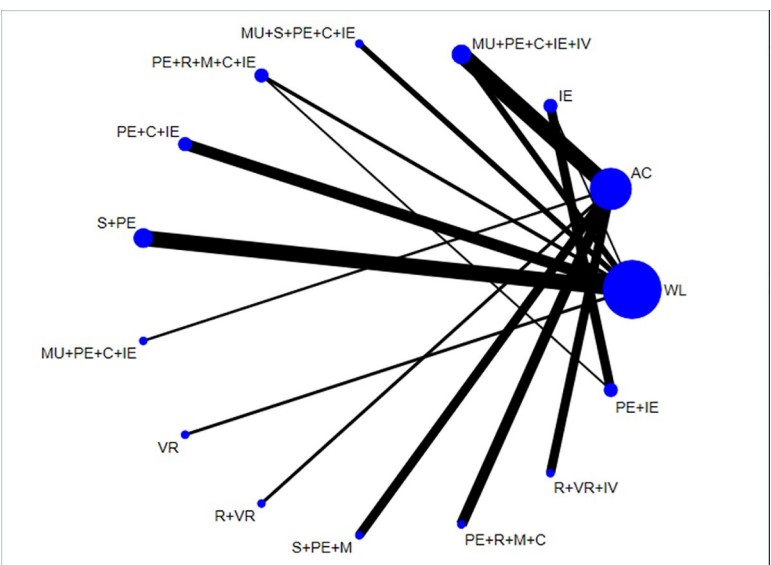

**Fig 9. Network diagram for all combinations of components extracted from included studies (edge thickness weighted by inverse variance).** AC, active control; C, cognitive restructuring; IE, imaginal exposure; IV, in vivo exposure; M, mindfulness; MU, multidimensional; PE, psychoeducation; R, relaxation; S, support; VR, virtual reality exposure; WL, waitlist.

There was insufficient data to explore interactions between multicomponent approaches and these intervention components.

## Discussion

The findings from this systematic review and meta-analysis suggest that collectively, psychological interventions are effective for treating PTSD symptoms, treating symptoms of common mental health problems, and improving sleep across all populations with a history of complex traumatic events. Evidence from non-RCTs generally supported this finding. These positive effects were especially pronounced for interventions with a trauma focus, such as TF-CBT and EMDR, and were observed over the longer term at 6 months and when compared with active controls. Non-trauma-focused interventions were not generally effective for PTSD symptoms, with only weak evidence in favour of IPT. There was less good evidence that psychological interventions were effective for managing the symptom cluster associated with disorders of self-regulation. We observed that TF-CBT was effective for managing negative self-concept and phase-based interventions were the leading candidate intervention to address interpersonal problems. No interventions were effective for managing emotional dysregulation. These findings were in the main endorsed by subgroup analyses across different populations exposed to complex traumatic events. In veteran and war-affected populations, TF-CBT and EMDR were associated with the greatest reductions in PTSD symptoms and symptoms of depression and anxiety, but there was a diminution in effect sizes when compared with the results from the pooled analyses across all populations. Similarly TF-CBT and EMDR were effective for reducing PTSD symptoms in refugees and populations exposed to childhood sexual abuse, although the precision of the treatment estimates was more uncertain in the analysis of childhood sexual abuse trials. The largest effect sizes were observed in the domestic violence subgroup analysis, which showed that TF-CBT was effective for managing PTSD symptoms, but this finding is based on limited evidence. The component NMA showed that multicomponent interventions that included at least cognitive restructuring and imaginal exposure were the

most effective for managing PTSD symptoms. Furthermore, analyses indicated that psychological interventions were associated with larger effect sizes than pharmacological interventions for managing PTSD symptoms, symptoms of depression, and sleep at posttreatment. Antipsychotics were shown to be effective for PTSD symptoms, but in the absence of safety data, our review does not offer findings that might overturn existing clinical practice guidelines that recommend against the use of risperidone [144]. Prazosin was the only other pharmacological therapy that conferred modest benefits for PTSD symptoms, and there is scope for revisiting recommendations against the use of this medication following further studies, especially in veterans.

These findings partly concur with Merz and colleagues, who recently showed that psychotherapeutic treatments are superior to pharmacological treatments for adults with PTSD at last follow-up but not at end of treatment, reaffirming the view that pharmacological therapy should not be used as first-line treatment for PTSD [145]. Our findings endorse this view and extend the relevance of international guideline recommendations that favour using TF-CBT and EMDR as first-line treatment for PTSD symptoms to those with histories of complex trauma.

When broken down by trauma exposure, we found a similar patterns of results observed in the pooled analyses across all populations. TF-CBT and EMDR were the most effective interventions for PTSD symptoms and common mental health problems for all subgroups. Heterogeneity was significantly reduced in the meta-analyses of the primary outcome for psychological interventions across all subgroups other than childhood sexual abuse. As previously shown, individual trauma-focused treatments are efficacious for adult survivors of childhood sexual abuse with PTSD, albeit analyses have so far failed to unpack which elements of trauma-focused interventions are most effective [146]. Furthermore, effectiveness of trauma-focused interventions can be reduced among the most complex cases of childhood sexual abuse with disturbances of self-organisation [147]. Similarly, previous reviews have shown that psychosocial interventions, and especially narrative exposure therapy, are effective for PTSD among refugees in both global and high-income settings [148, 149]. Although our findings show that trauma-focused interventions are also effective for mental comorbidities as well as PTSD among refugees, there are still uncertainties about how to practically address mental ill health among the unprecedented surge in refugees, especially in low-income settings [150].

Significantly, the size and durability of the treatment effects for PTSD and common mental health problems were diminished among veterans and war-affected populations when compared with the results from the pooled analyses across all populations. Veterans have high rates of mental comorbidity and experience high levels of problems that can negatively impact successful engagement with psychological treatment, such as interpersonal problems and emotional dysregulation [151]. Phase-based interventions that seek to address disturbances of self-organisation through skills-based strategies in combination with strategies that address traumatic memories were among the most promising therapeutic approaches for emotional dysregulation and interpersonal problems in veteran and childhood sexual abuse populations. TF-CBT was the most effective approach for managing negative self-concept. Using combinations of trauma-focused therapies and skills-based strategies in a flexible manner depending on symptom presentation is likely to be advantageous and removes the need for fixed approaches in cases of complex trauma [152].

This finding was partly endorsed by the component NMA, which showed that multicomponent interventions that included two or more intervention components are the most effective for managing PTSD symptoms in people with complex trauma. All effective multicomponent interventions included imaginal exposure and cognitive restructuring, but this superordinate group of interventions also included phase-based interventions that combined skills-based

strategies with trauma-focused strategies. In this sense, phase-based approaches can be realigned as multicomponent treatments with phasing conceptualised as an intervention component rather than a separate intervention category. There is emerging evidence that multicomponent interventions that can be delivered in an integrated or sequenced way and target more than one outcome are efficacious for people with multiple and often competing health and behavioural problems [153], including those with complex trauma [154].

Participants were less likely to drop out of psychological treatment than controls, even in studies judged to be at low risk of attrition bias, suggesting the difference in attrition between psychological intervention and controls is better explained by acceptability rather than attrition bias. Previously, it has been shown that dropout among active and ex-service military personnel is higher for TF-CBT than present-centred therapy, especially where prolonged exposure is used [155]. This has relevance for understanding how acceptability of interventions and patient preference can inform effective delivery of treatments for people with complex trauma. Patient preference for psychological interventions is commonly reported [156], but it is imperative that systems are put in place to ensure people's preferences are met to maximise likelihood of improving outcomes [157]. For example, we showed that mindfulness was an effective treatment for depression among veterans, but optimising delivery of such interventions as part of multicomponent packages needs to be cognisant of patient preferences about timing, setting, and format [158]. There is scope to explore how established evidence-based patient-centred frameworks such as the chronic care model can enhance and optimise the delivery of multicomponent care packages for people with complex trauma. Whereas there is ample evidence that multifaceted and collaborative care packages are effective for managing depression and chronic disease in primary care [159, 160], there is only limited evidence that such patient-centred care approaches are similarly effective for people with PTSD and mental health comorbidities [161].

Critical to any future research that might underpin patient-centred approaches is the need to capture outcomes that relate to broader notions about recovery that go beyond clinical recovery and include improvements in functioning and quality of life. We were only able to include data from six trials that measured quality of life, but it is well established that people with PTSD have profound deficits in quality of life and physical limitations, more so than people with other anxiety disorders [162]. This is especially true among populations exposed to complex trauma, such as veterans [163] and war-afflicted civilians [164], who often experience impairments across multiple life domains, including social and occupational functioning. Assessment of PTSD-related quality of life should therefore be a priority in the context of trials to improve the mental health of people exposed to complex trauma.

Additionally, it is important to go beyond assessment of PTSD symptoms and consider broader psychosocial difficulties that stem from the experience of complex traumatic events. This is especially true among refugee populations, whose emotional and behavioural problems are often linked to disruption in psychosocial systems that support mental health. Drawing on the Adaptation and Development After Persecution and Trauma (ADAPT) model, critical psychosocial systems include safety and security, interpersonal bonds and networks, justice, identities and roles, and existential meaning [165]. Treatment strategies that embrace the need to counter disruption to these psychosocial domains might prove effective for promoting a more positive refugee experience. A recent trial has shown that in refugees from Myanmar, a relatively brief 6-week course of integrative adapt therapy that is based on the ADAPT model led to improved adaptive capacity and resilience as well as greater reductions in PTSD symptoms and major depressive disorder compared with CBT [166]. Although the effect size for PTSD symptoms was smaller in this trial than those reported in our meta-analyses of psychological interventions among refugees, it might be that supporting adaptation to the refugee experience is as important as symptom control.

## Strengths and limitations

This review attempted to capture the totality of all controlled evidence about the effectiveness of psychological and pharmacological treatments for people exposed to complex trauma. We included non-RCTs on the basis that these studies might include data about novel treatments delivered in pragmatic settings, but the evidence from these trials was eclipsed by the evidence from randomised comparisons, which offered the most robust assessments of treatment effectiveness. Our review has a number of strengths that further enhance the robustness of the findings. By taking an approach that favoured inclusion based on trauma exposure rather than diagnosis, we were able to develop and operationalise broad inclusion criteria for the population of interest. In doing so, our search was not tied to a narrowly defined group of studies that exclusively evaluated interventions in populations with the as yet empirically untested diagnostic label of CPTSD, but rather captured a broader set of studies that addressed mental health problems in people exposed to complex traumatic events.

Additional strengths of the review include the application of component NMA approaches to understanding treatment effectiveness and moderators of effectiveness. By searching extensively and adopting a broad approach to inclusion, we were able to assemble a much larger data set than in previous reviews, enhancing our ability to quantify and explore heterogeneity and—for the first time, to our knowledge—disentangle the effects of individual components of composite interventions. NMA offers additional benefits over standard pairwise analyses in that the comparative efficacy of specific interventions can be estimated and ranked, even when two treatments have never been compared directly head-to-head. Furthermore, since NMA can improve the precision of estimates by allowing integration of both direct and indirect treatment effect estimates, it is recommended over pairwise meta-analyses by the World Health Organization as a basis for clinical guidelines [167].

Despite using an extensive search strategy and applying broad inclusion criteria, our review has an underrepresentation of studies with a focus on complex-trauma populations drawn from prison settings and survivors of torture and forced migrant labour, otherwise known as modern slavery. Future work should look to identify ways to ensure these populations are not overlooked. In addition, our search did not capture a critical mass of studies that included outcomes related to comorbid psychiatric states such as borderline personality disorder. This might have been offset had we adopted a more clinical and diagnostic approach to our inclusion criteria. Whereas we did include populations with comorbidities, including psychosis and common mental health problems, we excluded those with dual diagnosis of complex trauma and substance and alcohol misuse on the grounds that these populations are likely to require care that is different from and more specialist than that typically provided in the context of PTSD. However, recent work has shown that treatment-seeking veterans are more likely to report alcohol dependence and alcohol harm than active military personnel or the general population, highlighting the need in the future to assess the efficacy of mental health interventions for complex-trauma populations with specific needs [168].

Benefits of treatment can diminish over the longer term, especially in populations exposed to complex trauma. However, most trials included in this review only reported posttreatment and short-term outcomes, limiting evaluation of medium- and longer-term outcomes. People with complex-trauma experiences can benefit over the longer term from psychological therapies, but higher levels of mental health comorbidities are associated with poorer PTSD treatment response [169], suggesting that measurement of important secondary outcomes as well as PTSD symptoms is critical to understanding longer-term impact of treatments.

There was consistent evidence for the effectiveness of several psychological interventions, especially TF-CBT and EMDR, for improving PTSD, depression, and anxiety symptoms. Effect

estimates were lower for pharmacological interventions and lacked precision. However, we did not make any formal comparisons between psychological and pharmacological interventions either based on direct comparisons in trials or through NMAs, and as such, any informal comparisons are inherently uncertain. Furthermore, it could be argued that comparisons about findings from RCTs of psychological versus pharmacological interventions might favour the former, where blinding may be absent and a control for attention is missing. However, there is compelling meta-epidemiological evidence that estimated treatment effects do not differ between trials with and without blinding of patients, healthcare providers, or outcome assessors [170].

Although we were able to judge the acceptability of interventions, there was insufficient data to assess harms related to either psychological or pharmacological interventions. Harms go beyond negative outcomes and refer to enduring negative effects that are directly caused by the therapy. The absence of harms data is more prevalent for psychological trials than pharmacological trials [171], and this is an important omission given that at least 1 in 20 people report lasting bad effects from psychological treatment [172]. Going forwards, there is a solid case to collect quantitative data about adverse events and clinically significant worsening of symptoms during and shortly after treatment, and also qualitative data about patients experience of harm [173].

The NMA methods used were robust for most intervention components, but credible intervals were wide, indicating very imprecise estimates. This reflects the exploratory nature of the analyses in which we assessed a number of covariates. In addition, there were insufficient studies to tease apart the relative contribution of skills-based components, and these were pragmatically classed as multicomponent interventions. Finally, most studies included in the NMA had small sample sizes and high heterogeneity and were rated at either moderate or high risk of bias. Therefore, all estimates should be interpreted cautiously.

## Conclusion

In conclusion, existing evidence-based psychological trauma-focused interventions are effective for managing PTSD symptoms and mental health comorbidities in people with complex-trauma histories. There was less good evidence that pharmacological interventions were effective for PTSD or mental health comorbidities in the presence of complex-trauma exposure. Trauma-focused interventions were generally less effective for managing disturbances of self-organisation as per ICD-11 definitions, with multicomponent interventions showing some promise for managing these symptom clusters. Overall, multicomponent interventions that included at least imaginal exposure and cognitive restructuring were the most effective for managing PTSD symptoms in complex trauma. There is a case for reconceptualising phasing as an element of multicomponent interventions and for the focus of the research and clinical community to now develop efficient and effective patient-centred strategies for delivery of multicomponent treatments for complex trauma.

## Supporting information

**S1 Text. PRISMA NMA checklist of items to include when reporting a systematic review involving an NMA.** NMA, network meta-analysis.
(DOCX)

**S2 Text. Sample search strategy in Ovid MEDLINE.**
(DOCX)

**S1 Table. Characteristics of included studies.** ACT, acceptance and commitment therapy; CBT, cognitive behavioural therapy; DBT, dialectical behavioural therapy; EMDR, eye movement desensitisation and reprocessing; IPT, interpersonal therapy; NTCBT, non-trauma-focused CBT; MBCT, mindfulness-based cognitive therapy; MBSR, mindfulness-based stress reduction; PE, prolonged exposure; NR, not reported; RCT, randomised controlled trial; SSRI, selective serotonin reuptake inhibitor; STAIR, skills training in affective and interpersonal regulation; TAU, treatment as usual; TFCBT, trauma-focused CBT.
(DOCX)

**S2 Table. Risk of bias assessments for randomised controlled trials.**
(DOCX)

**S3 Table. Risk of bias assessments for non–randomised controlled trial.** -, significant sources of bias; +, potential sources of bias; ++, minimal sources of bias; NA, not applicable; NR, not reported.
(DOCX)

**S4 Table. Effect sizes (standardised mean difference) for psychological and pharmacological interventions versus control in all populations.** BDI, Beck depression inventory; CAPS, clinician-administered PTSD scale; CBT, cognitive behavioural therapy; EMDR, eye movement desensitisation and reprocessing therapy; IPT, interpersonal therapy; PANSS, positive and negative syndrome scale; PTSD, posttraumatic stress disorder; SSRI, selective serotonin reuptake inhibitor; TF-CBT, trauma-focused cognitive behavioural therapy.
(DOCX)

**S5 Table. Effect sizes (standardised mean difference) for psychological interventions versus control for complex-trauma exposure subgroups.** EMDR, eye movement desensitisation and reprocessing therapy; PTSD, posttraumatic stress disorder; TF-CBT, trauma-focused cognitive behavioural therapy.
(DOCX)

**S6 Table. Mean difference for outcomes by intervention component.**
(DOCX)

## Author Contributions

**Conceptualization:** Peter A. Coventry, Melanie Temple, Marylène Cloitre, Thanos Karatzias, Jonathan Bisson, Neil P. Roberts, Corrado Barbui, Rachel Churchill, Simon Gilbody.

**Data curation:** Peter A. Coventry, Nick Meader.

**Formal analysis:** Nick Meader, Hollie Melton, Holly Dale, Kath Wright, Jennifer V. E. Brown.

**Funding acquisition:** Peter A. Coventry, Corrado Barbui, Rachel Churchill, Karina Lovell, Dean McMillan, Simon Gilbody.

**Investigation:** Peter A. Coventry.

**Methodology:** Nick Meader.

**Project administration:** Peter A. Coventry, Hollie Melton.

**Software:** Kath Wright.

**Supervision:** Peter A. Coventry.

**Writing – original draft:** Peter A. Coventry.

**Writing – review & editing:** Peter A. Coventry, Nick Meader, Hollie Melton, Melanie Temple, Holly Dale, Kath Wright, Marylène Cloitre, Thanos Karatzias, Jonathan Bisson, Neil P. Roberts, Jennifer V. E. Brown, Corrado Barbui, Rachel Churchill, Karina Lovell, Dean McMillan, Simon Gilbody.

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
