## [Editor Report · Decision Letter 0]

11 Mar 2020

Dear Dr Coventry, 

Thank you for submitting your manuscript entitled "Psychological and pharmacological interventions for PTSD and comorbid mental health problems following complex traumatic events: systematic review and component network meta-analysis" for consideration by PLOS Medicine.

Your manuscript has now been evaluated by the PLOS Medicine editorial staff and I am writing to let you know that we would like to send your submission out for external peer review.

Kind regards,

Helen Howard, for Clare Stone PhD 

Acting Editor-in-Chief

PLOS Medicine 

plosmedicine.org

---

## [Decision Letter · Decision Letter 1]

23 Apr 2020

Dear Dr. Coventry,

Thank you very much for submitting your manuscript "Psychological and pharmacological interventions for PTSD and comorbid mental health problems following complex traumatic events: systematic review and component network meta-analysis" (PMEDICINE-D-20-00773R1) for consideration at PLOS Medicine. 

[LINK]

In light of these reviews, I am afraid that we will not be able to accept the manuscript for publication in the journal in its current form, but we would like to consider a revised version that addresses the reviewers' and editors' comments. Obviously we cannot make any decision about publication until we have seen the revised manuscript and your response, and we plan to seek re-review by one or more of the reviewers. 

We expect to receive your revised manuscript by May 14 2020 11:59PM. Please email us (plosmedicine@plos.org) if you have any questions or concerns.

We look forward to receiving your revised manuscript. 

Sincerely,

Emma Veitch, PhD

PLOS Medicine

On behalf of Clare Stone, PhD, Acting Chief Editor,

PLOS Medicine

plosmedicine.org

*In the last sentence of the Abstract Methods and Findings section, please describe the main limitation(s) of the study's methodology.

*At this stage, we ask that you include a short, non-technical Author Summary of your research to make findings accessible to a wide audience that includes both scientists and non-scientists. The Author Summary should immediately follow the Abstract in your revised manuscript. This text is subject to editorial change and should be distinct from the scientific abstract. Please see our author guidelines for more information: https://journals.plos.org/plosmedicine/s/revising-your-manuscript#loc-author-summary

*The reviewers have raised the issue of whether it's possible to report on any harms (adverse effects) findings of the included studies, or to highlight/elaborate on the absence of this data in the included trials, if relevant - the editors would agree with this point and suggest it be covered in some way depending on what the data can support.

*Reviewers have also queried the inclusion of non-randomized data in the meta-analyses. Obviously it is up to the authors to decide how to address this point but some sensitivity analysis around this issue might be merited - ie to note how the main conclusions might change if only based on randomized trial data (which would be more conventional). 

Comments from the reviewers:

Reviewer #1: I confine my remarks to statistical aspects of this paper. Mostly, they were fine, but I have a few points that need clarification before I can recommend publication.

General: Network meta analysis relies on the assumption of exchangeability. How was this tested? I did not see it.

Lines 260-261 and 267 Please give some detail. 

Plots: What are the bars around the points? Is that scaled to weight, or what?

Peter Flom

Reviewer #2: Many thanks for inviting me to review this systematic review. It is most comprehensive with rigorous analysis on rich data, to identify evidence for effective treatment for a complex condition.

I would like to propose the following, for the authors' consideration in further revising/developing the paper:

1. Further justifications in including non-randomised trials in this SR is indicated - given there are sufficient data from RCTs as evidenced in the meta-analyses and NMA, the value of non-randomised trials is doubtful. Including effect sizes from a single non-randomised trials in the results of secondary outcomes and subgroup analyses, in my view, does not add to the evidence.

2. In reporting the results, please provide an overview of the studies and the interventions investigated. It is of essence to understand the nature and intensity of the interventions, e.g. clinical settings, programme comprising xx sessions? range of treatment duration (e.g. over 6 months), time/dose of cumulative sessions, .... etc. Same applies to the pharmacological interventions. Such a summary description will illustrate the intervention intensity and design better for the readers to interpret the results of their effectiveness. It will also set the scene for the discussion arguing phasing being an element of multi-component interventions targeting a range of symptoms in addition to core PTSD symptoms in this population.

3. In reporting the results, it is preferable for all the statistically significant MAs to be written out in full, i.e. k=xx, n=xxx, SMD, 95% CI, p value, I squared, as they help the readers to understand the effects the authors report through texts and to interpret the effects in the context of the available evidence. Also in reporting the primary analysis pulling all psychological interventions comparing with all controls across all populations, think it is better that the results are reported separately for post-treatment (plus the point above re good to know how long treatment lasted on average), and at other follow-up time points (think only 6-months).

4. In terms of limitations, I would suggest commenting on the lack of medium or long-term follow up data.

5. Please check through the numbers of excluded full text in the PRISMA flowchart: the numbers do not add up to 402.

6. Please also check through the captions for Figures 4-6.

Reviewer #3: This is an extremely important meta-analysis, extending knowledge of interventions for PTSD following complex trauma, and addressing a major point of contention in PTSD treatment research (de Jong, et al, 2016). This contention has been largely argued from evidence from single studies and clinical opinion so the addition of large-scale empirical evidence is critical and timely. The review not only addresses this issue but also presents a re-conceptualisation of the binary between phased-based and non-phased-based approaches, which will progress thinking in PTSD treatment. 

My expertise in not in meta-analysis methodology so I will leave that to other reviewers to consider. 

I couldn't find the results on harms (worsening of PTSD).

I was struck by the few studies in the meta-analysis that had measured quality of life and the lack of statistically significant improvement in those few studies that had measured it. Considering the movement to broaden the conceptualisation of, and include patient perspectives on, outcomes (the recovery movement), would the authors consider making some kind of statement about quality of life results in the discussion?

[LINK]

---

## [Decision Letter · Decision Letter 2]

24 Jun 2020

Dear Dr. Coventry,

Thank you very much for re-submitting your manuscript "Psychological and pharmacological interventions for PTSD and comorbid mental health problems following complex traumatic events: systematic review and component network meta-analysis" (PMEDICINE-D-20-00773R2) for review by PLOS Medicine.

I have discussed the paper with my colleagues and the academic editor and it was also seen again by one of the original reviewers. I am pleased to say that provided the remaining editorial and production issues are dealt with we are planning to accept the paper for publication in the journal.

[LINK]

We look forward to receiving the revised manuscript by Jul 01 2020 11:59PM. 

Sincerely,

Thomas McBride, PhD

Senior Editor 

PLOS Medicine

plosmedicine.org

Requests from Editors:

1- In the Abstract Background, please provide more context of why the study is important. Some of the points made in the first section of your Author Summary would be appropriate here, but please rephrase a bit. 

2- In addition, the Abstract begins with “There has never been a meta-analysis of the effectiveness ...” while the Discussion begins with “This is the most comprehensive and wide ranging... ”, which seem incompatible. Please ensure these claims are consistent. Please either remove the sentence beginning with “There has never been a meta-analysis of the effectiveness ...” or add “To our knowledge…” or similar. 

3- The final sentence of the Abstract Background should clearly state the study question.

4- In the Abstract Methods and Findings section, please include the settings included, average length of follow up, and a demographic breakdown of the populations included.

5- "in people" at line 41 (of the marked up manuscript).

6- Please add a brief summary of the secondary endpoints at line 45 after the primary endpoint is quoted. 

7- Please include the findings for pharmacological interventions (from line 440) in the Abstract Methods and Findings section.

8- In the Abstract Methods and Findings section, please quantify the results for all outcomes (including 95% CIs and p-values).

9- In the Abstract Conclusions, please add the phrase "In this study, we observed ..." or similar, to avoid overreaching what can be concluded from the data.

10- Thank you for adding an Author Summary. Please remove the phrase “...most wide-ranging and comprehensive…” from the 5th point.

11- Thank you for including your PRISMA checklist. Please replace the page numbers with paragraph numbers per section (e.g. "Methods, paragraph 1"), since the page numbers of the final published paper may be different from the page numbers in the current manuscript.

12- Thank you for noting the process for including non-randomized studies in the Methods section. Please clarify the independent peer review was during the protocol development, to differentiate from the peer review at this journal.

13- Please update your search to the present time.

14- Please include a bit more information in the Results on the studies that were included in the systematic review but not in the meta-analysis.

15- Should the results for the pharmacological interventions be included in a figure?

16- Please provide more details in the figure legends, including spelling out any abbreviations used (even if they were described in the main text), and describing the black bars vs the grey shading (95% CIs and weight, respectively?). 

17- In addition to the effect sizes reported here, it would be useful to translate these effects into more interpretable measures, such as number or severity of symptoms and measures of daily functioning and quality of life.

18- Though you report the improvements for quality of life and positive and negative affect measures were not significant, please still quantify these in the text.

19- Please restructure the early part of the discussion section so that the first paragraph provides a summary of the study's findings

20- It can perhaps be argued that comparisons about findings from RCTs of psychological versus pharmaceutical treatments might favour the former, where blinding may be absent and a control for attention missing. You may wish to comment on this question in the paragraph on limitations in your discussion section. 

21- Please rephrase the language in the first paragraph of the Discussion to remove “comprehensive and wide ranging” and “novel approaches”. 

22- At line 633 (marked up document), please edit to “Results from the pairwise meta-analyses suggest that psychological interventions are effective…”

23- At line 648, you note that your results suggest psychotherapeutic treatments are superior to pharmacological treatments for PTSD, were these based on direct or indirect comparisons? If the latter, perhaps this language should be toned down and this limitation also should be noted in the Strengths and Limitations section.

24- Line 734, remove “robust and innovative”.

25- Please remove the sentence at lines 743-745.

26- The wording of the sentence starting on line 746 is a bit confusing. Perhaps: “Despite using an extensive search strategy and applying broad inclusion criteria, our review has an underrepresentation of studies with a focus on complex trauma populations drawn from prison settings and survivors of torture and forced migrant labour, otherwise known as modern slavery.”

27- Thank you for including the overall risk of bias assesment in table S1. Please also include a separate table breaking down risk for each study by individual components. 

28- It looks like a formatting error in reference 18, please correct.

29- DOIs are missing for several references, please update.

30- Please update the Data statement to note that data are available from the primary research papers, which are listed in the references.

Comments from Reviewers:

Reviewer #3: The authors have dealt with my comments to my satisfaction.

[LINK]

---

## [Editor Report · Decision Letter 3]

15 Jul 2020

Dear Dr Coventry, 

On behalf of my colleagues and the academic editor, Dr. Sarah Bendall, I am delighted to inform you that your manuscript entitled "Psychological and pharmacological interventions for post-traumatic stress disorder and comorbid mental health problems following complex traumatic events: systematic review and component network meta-analysis" (PMEDICINE-D-20-00773R3) has been accepted for publication in PLOS Medicine. 

PRODUCTION PROCESS

PRESS

PROFILE INFORMATION

Thank you again for submitting the manuscript to PLOS Medicine. We look forward to publishing it. 

Best wishes, 

Thomas McBride, PhD

Senior Editor 

PLOS Medicine

plosmedicine.org